# 2DQuant: Low-bit Post-Training Quantization for Image Super-Resolution

**Kai Liu**[1], **Haotong Qin**[2], **Yong Guo**[3], **Xin Yuan**[4],
**Linghe Kong**[1*], **Guihai Chen**[1], **Yulun Zhang**[1*]
[1]Shanghai Jiao Tong University, [2]ETH Zürich,
[3]Max Planck Institute for Informatics, [4]Westlake University

## Abstract

Low-bit quantization has become widespread for compressing image super-resolution (SR) models for edge deployment, which allows advanced SR models to enjoy compact low-bit parameters and efficient integer/bitwise constructions for storage compression and inference acceleration, respectively. However, it is notorious that low-bit quantization degrades the accuracy of SR models compared to their full-precision (FP) counterparts. Despite several efforts to alleviate the degradation, the transformer-based SR model still suffers severe degradation due to its distinctive activation distribution. In this work, we present a dual-stage low-bit post-training quantization (PTQ) method for image super-resolution, namely **2DQuant**, which achieves efficient and accurate SR under low-bit quantization. The proposed method first investigates the weight and activation and finds that the distribution is characterized by coexisting symmetry and asymmetry, long tails. Specifically, we propose Distribution-Oriented Bound Initialization (DOBI), using different searching strategies to search a coarse bound for quantizers. To obtain refined quantizer parameters, we further propose Distillation Quantization Calibration (DQC), which employs a distillation approach to make the quantized model learn from its FP counterpart. Through extensive experiments on different bits and scaling factors, the performance of DOBI can reach the state-of-the-art (SOTA) while after stage two, our method surpasses existing PTQ in both metrics and visual effects. 2DQuant gains an increase in PSNR as high as 4.52dB on Set5 ($\times$2) compared with SOTA when quantized to 2-bit and enjoys a 3.60$\times$ compression ratio and 5.08$\times$ speedup ratio. The code and models are available at https://github.com/Kai-Liu001/2DQuant.

## 1 Introduction

As one of the most classical low-level computer vision tasks, image super-resolution (SR) has been widely studied with the significant development of deep neural networks. With the ability to reconstruct high-resolution (HR) image from the corresponding low-resolution (LR) image, SR has been widely used in many real-world scenarios, including medical imaging [13, 21, 19], surveillance [46, 39], remote sensing [1], and mobile phone photography. With massive parameters, DNN-based SR models always require expensive storage and computation in the actual application. Some works have been proposed to reduce the demand for computational power of SE models, like lightweight architecture design and compression. One kind of approach investigates lightweight and efficient models as the backbone for image SR. This progression has moved from the earliest convolutional neural network (CNNs) [10, 11, 25, 49] to Transformers [48, 32, 44, 42, 4, 3] and their combinations. The parameter number significantly decreased while maintaining or even enhancing performance. The other kind of approach is compression, which focuses on reducing the parameter (*e.g.*, pruning and distillation) or bit-width (quantization) of existing SR models.

Model quantization [7, 9, 20, 31] is a technology that compresses the floating-point parameters of a neural network into lower bit-width. The discretized parameters are homogenized into restricted

---

*Corresponding authors: Yulun Zhang, yulun100@gmail.com, Linghe Kong, linghe.kong@sjtu.edu.cn

38th Conference on Neural Information Processing Systems (NeurIPS 2024).

candidate values and cause heterogenization between the FP and quantized models, leading to severe performance degradation. Considering the process, quantization approaches can be divided into quantization-aware training (QAT) and post-training quantization (PTQ). QAT simultaneously optimizes the model parameters and the quantizer parameters [6, 16, 26, 50], allowing them to adapt mutually, thereby more effectively alleviating the degradation caused by quantization. However, QAT often suffers from a heavy training cost and a long training time, and the burden is even much heavier than the training process of the FP counterparts, which necessitates a large amount of compatibility and makes it still far from practical in training-resource-limited scenarios.

Fortunately, post-training quantization emerges as a promising way to quantize models at a low training cost. PTQ fixes the model parameters and only determines the quantizer parameters through search or optimization. Previous researches [41, 26] on PTQ for SR has primarily focused on CNN-based models such as EDSR [33] and SRResNet [24]. However, these quantization methods are

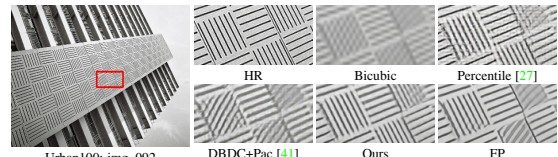

HR    Bicubic    Percentile [27]

DBDC+Pac [41]    Ours    FP

Urban100: img_092

Figure 1: Existing methods suffer from blurring artifacts.

not practical for deployment for two reasons. **Firstly**, these CNN-based models themself require huge space and calculation resources. Their poor starting point makes them inferior to advanced models in terms of parameters and computational cost, even after quantization. As shown in Table 1, the light version of SwinIR needs only 16.2% parameters and 15.9% FLOPs compared with quantized EDSR. But its PSNR metric is close to that of the FP EDSR. While the previous PTQ algorithm, DBDC+Pac, suffers from unacceptable degradation in both visual and metrics. **Secondly**, most of these methods can not adapt well to Transformer-based models because of the deterioration of self-attention in quantized transformers. As shown in Figure 1, when applied on SwinIR, the existing methods still suffer from distorted artifacts compared with FP or HR.

Therefore, we conducted a post-training quantization analysis on super-resolution with a classical Transformer-based model SwinIR [32]. The weight and activation

Table 1: Complexity and performance ($\times 4$).

| Model | EDSR [33] | EDSR (4bit) [41] | SwinIR-light [32] | DBDC+Pac (4bit) [41] | Ours (4bit) |
|---|---|---|---|---|---|
| Params (MB) | 172.36 | 21.55 | 3.42 | 1.17 | 1.17 |
| Ops (G) | 823.34 | 103.05 | 16.74 | 4.19 | 4.19 |
| PNSR on Urban100 | 26.64 | 25.56 | 26.47 | 24.94 | 25.71 |

distribution is characterized by coexisting symmetry and asymmetry, long tails. Firstly, if the previous symmetric quantization method is applied for asymmetric distribution, at least half of the candidates are completely ineffective. Besides, the long tail effect causes the vast majority of floating-point numbers to be compressed into one or two candidates, leading to worse parameter homogenization. Furthermore, with such a small number of parameters, SwinIR's information has been highly compressed, and quantizing the model often results in significant performance degradation. Nevertheless, the excellent performance and extremely low computational requirements of Transformer-based models are precisely what is needed for deployment in real-world scenarios.

In this paper, we propose **2DQuant**, a two-stage PTQ algorithm for image super-resolution tasks. To enhance the representational capacity in asymmetry scenarios, we employ a quantization method with two bounds. The bounds decide the candidate for numbers out of range and the interval of candidates in range. **First**, we propose **distribution-oriented Bound Initialization** (DOBI), a fast MSE-based searching method. It is designed to minimize the value heterogenization between quantized and FP models. Two different MSE [5] search strategies are applied for different distributions to avoid nonsense traversal. This guarantees minimum value shift while maintaining high speed and efficiency in the search process. **Second**, we propose **Distillation Quantization Calibration** (DQC), a training-based method. It is designed to adjust each bound to its best position finely. This ensures that the outputs and intermediate feature layers of the quantized model and that of the FP model should remain as consistent as possible. Thereby DQC allows the quantizer parameters to be finely optimized toward the task goal. The contributions of this paper can be summarized as follows:

(1) To the best of our knowledge, we are the first to explore PTQ with Transformer-based model in SR thoroughly. We design 2DQuant, a unique and efficient two-stage PTQ method (see Figure 2) for image super-resolution, which utilizes DOBI and DQC to optimize the bound from coarse to fine.

(2) In the first stage of post-quantization, we use DOBI to search for quantizer parameters, employing customized search strategies for different distributions to balance speed and accuracy. In the second stage, we design DQC, a more fine-grained optimization-based training strategy, for the quantized model, ensuring it aligns with the FP model on the calibration set.

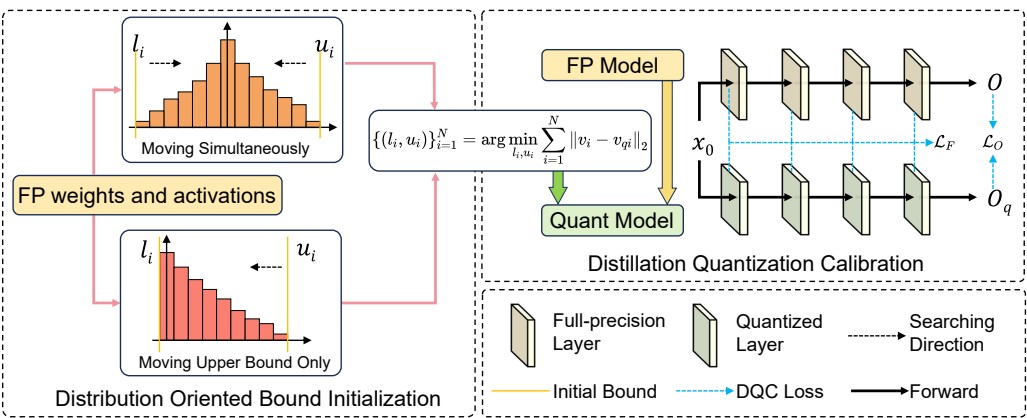

Figure 2: The overall pipeline of our proposed 2DQuant method. The whole pipeline contains two stages, optimizing the clipping bound from coarse to fine. In stage 1, we design DOBI to efficiently obtain the coarse bound. In stage 2, DQC is performed to finetune clipping bounds and guarantee the quantized model learns the full-precision (FP) model's feature and output information.

(3) Our 2DQuant can compress Transformer-based model to 4,3,2 bits with the compression ratio being $3.07\times$, $3.31\times$, and $3.60\times$ and speedup ratio being $3.99\times$, $4.47\times$, and $5.08\times$. No additional module is added so 2DQuant enjoys the theoretical upper limit of compression and speedup.

(4) Through extensive experiments, our 2DQuant surpasses existing SOTA on all benchmarks. We gain an increase in PSNR by as high as 4.52dB in Set5 ($\times2$) when compressed to 2 bits, and our method has a more significant increase when compressed to lower bits.

## 2  Related work

**Image super-resolution.**   Deep CNN networks have shown excellent performance in the field of image super-resolution. The earliest SR-CNN [10, 11] method adopted a CNN architecture. It surpassed previous methods in the image super-resolution domain. In 2017, EDSR [33] won the NTIRE2017 [40] championship, becoming a representative work of CNNs in the SR by its excellent performance. Thereafter, with the continuous development of Vision Transformers (ViT) [12], models based on the ViT architecture have surpassed many CNN networks. These Transformer-based models achieve significant performance improvements and they have fewer parameters and lower computational costs. Many works have modified the ViT architecture, achieving continuous improvements. A notable example is SwinIR [32]. With a simple structure, it outperforms many CNN-based models. However, previous explorations of post-quantization in the super-resolution domain have been limited to CNN-based models. They focus on models like EDSR [33] or SRResNet [24]. It is a far cry from advanced models no matter in parameters, FLOPs, or performance. Currently, there is still a research gap in post-quantization for Transformer architectures.

**Model quantization.**   Model quantization is used to compress large models [30, 28, 29] and can be divided into QAT and PTQ. QAT is widely accepted due to its minimal performance degradation. PAMS [26] utilizes a trainable truncated parameter to dynamically determine the upper limit of the quantization range. DAQ [17] proposed a channel-wise distribution-aware quantization scheme. CADyQ [16] is designed for SR networks and optimizes the bit allocation for local regions and layers in the input image. However, QAT usually requires training for as long as or even longer than the original model, which becomes a barrier for real scenarios deployment. Instead of training the model from scratch, existing PTQ methods use the pre-trained models. PTQ algorithms only find the clipping bound for quantizers, saving time and costs. DBDC+Pac [41] is the first to optimize the post-training quantization for image super-resolution task. It outperforms other existing PTQ algorithms. Whereas, they only focus on EDSR [33] and SRResNet [24]. Their 4-bit quantized version is inferior to advanced models in terms of parameters and computational cost, let alone performance. It reveals a promising result for PTQ on SR, and using a more advanced model could bridge the gap between high-performance models and limited calculation resource scenarios.

## 3  Methodology

To simulate the precision loss caused by quantization, we use fake-quantize [22], *i.e.* quantization-dequantization, for activations and weights. and the process can be written as

$$v_c = \text{Clip}(v, l, u), \quad v_r = \text{Round}(\frac{2^N - 1}{u - l}(v_c - l)), \quad v_q = \frac{u - l}{2^N - 1}v_r + l, \tag{1}$$

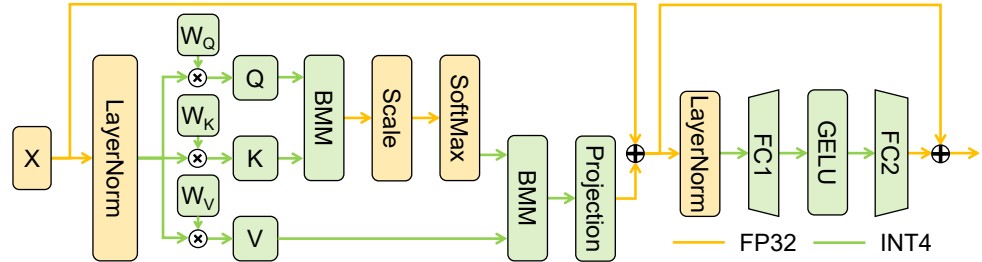

Figure 3: Quantization scheme for SwinIR Transformer blocks. Fake quantization and INT arithmetic are performed in all compute-intensive operators including all linear layers and batch matmul. Lower bits such as 3 or even 2 are also permitted. Dropout of attention and projection is ignored

where $v$ denotes the value being fake quantized, which can be weight or activation. $l$ and $u$ are the lower and upper bounds for clipping, respectively. $\text{Clip}(v, l, u) = \max(\min(v, u), l)$, and Round rounds the input value to the nearest integer. $v_c$ denotes the value after clipping, and $v_r$ denotes the integer representation of $v$, and $v_q$ denotes the value after fake quantization. The Clip and Round operations contribute to reducing the parameters and FLOPs but also introduce quantization errors.

Figure 3 shows the basic structure of the Transformer block. We have quantized all the modules with a significant computational load within them, effectively reducing the model's FLOPs. Table 2 shows the FLOPs needed for each module. The Linear layers and matrix multiplication account for approximately 86% of the computation load, which are all transformed into integer arithmetic. When

Table 2: FLOPs distribution.

| Module | FLOPs (G) | Ratio (%) |
|---|---|---|
| Linear & BMM | 14.34 | 85.66 |
| Conv | 2.33 | 13.90 |
| Other | 0.07 | 0.44 |
| Total | 16.74 | 100.00 |

performing gradient backpropagation, we follow the Straight-Through Estimator [8] (STE) style:

$$\frac{\partial v_q}{\partial u} = \frac{\partial v_c}{\partial u} + \frac{1}{2^n - 1} v_r - \frac{v_c - l}{u - l}, \quad \frac{\partial v_q}{\partial l} = \frac{\partial v_c}{\partial l} - \frac{1}{2^n - 1} v_r + \frac{v_c - l}{u - l}, \quad (2)$$

where $\frac{\partial v_c}{\partial u} = H(u - v)$ and $\frac{v_c}{\partial l} = H(l - v)$, $H(\cdot)$ denotes Heaviside step function [47]. This formula approximates the direction of gradient backpropagation, allowing training-based optimization to proceed. The derivation of the formula can be found in the supplementary material.

Figure 2 shows the whole pipeline of 2DQuant, which is a **two**-stage coarse-to-fine post-training quantization method. The first stage is **D**OBI, using **two** strategies to minimize the value shift while the second stage is **D**QC, optimizing **two** bound of each quantizer towards the task goal.

## 3.1 Analysis of data distribution

To achieve better quantization results, we need to analyze the distribution of the model's weights and activations in detail. We notice that the data distribution shows a significantly different pattern from previous explorations, invalidating many of the previous methods. The weights and activations distribution of SwinIR is shown in Figure 4. More can be found in supplemental material. Specifically, the weights and activations of SwinIR exhibit noticeable long-tail, coexisting symmetry and asymmetry.

**Weight.** The weights of all linear layers are symmetrically distributed around zero, showing clear symmetry, and are generally similar to a normal distribution. This is attributed to the weight decay applied to weights, which provides quantization-friendly distributions. From the value shift perspective, both symmetric and asymmetric quantization are tolerable. Whereas, from the vantage point of task objectives, asymmetric quantization possesses the potential to offer a markedly enhanced information density, thus elevating the overall precision of the computational processes involved.

**Activations.** As for activations, they exhibit obvious periodicity in different Transformer Blocks. For V or the input of FC1, the obtained activation values are symmetrically distributed around 0. However, for the attention map or the input of FC2 in each Transformer Block, due to the Softmax calculation or the GELU [14] activation function, the minimum value is almost fixed, and the overall distribution is similar to an exponential distribution. Therefore, the data in SwinIR's weights and activations exhibit two distinctly different distribution characteristics. Setting asymmetric quantization and different search strategies for both can make the search rapid and accurate.

## 3.2 Distribution-oriented bound initialization

Because the data distribution exhibits a significant long-tail effect, we must first clip the range to avoid low effective bits. Common clipping methods include density-based, ratio-based, and MSE-based approaches. The first two require manually specifying the clipping ratio, which significantly affects the clipping outcome and necessitates numerous experiments to determine the optimal ratio. Thus we proposed the Distribution-Oriented Bound Initialization (DOBI) to search the bound for weight and

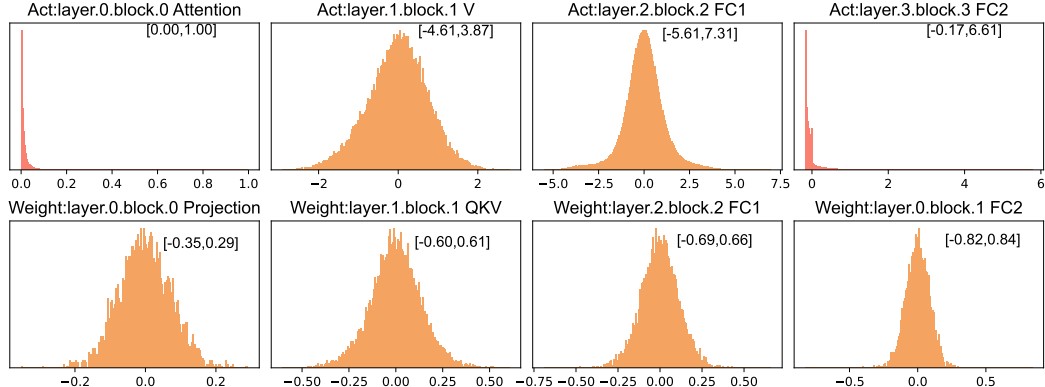

Figure 4: The selected representative distribution of activations (Row 1) and weights (Row 2). The range of data is marked in the figure. All weights obey symmetric distribution. The attention map and the input of FC2 are asymmetric due to softmax function and GELU function.

activation, avoiding manually adjusting hyperparameters. The global optimizing goal is as follows

$$\{(l_i, u_i)\}_{i=1}^N = \arg\min_{l_i, u_i} \sum_{i=1}^N \|v_i - v_{qi}\|_2 . \tag{3}$$

The collection of all quantizers' bounds $\{(l_i, u_i)\}_{i=1}^N$ is the linchpin of quantized model performance as it indicates the candidate value for weights and activations. We note that the data distribution falls into two categories: one resembling a bell-shaped distribution and the other resembling an exponential distribution. For the bell-shaped distribution, we use a symmetric boundary-narrowing search method. Whereas, for the exponential distribution, we fix the lower bound to the minimum value of the data and only traverse the right bound. The specific search method is shown in Algorithm 1. The time complexity of Algorithm 1 is $\mathcal{O}(MK)$, where $M$ is the number of elements in data $v$ and $K$ is the number of search points. The condition $v$ *is symmetrical* is obtained by observing the visualization of $v$ and the activations are from the statistics on a small calibration set.

### 3.3 Distillation quantization calibration

To further fine-tune the clipping range, we propose distillation quantization calibration (DQC) to transfer the knowledge from the FP model to the quantized model. It leverages the knowledge distillation [15] where the FP model acts as the teacher while the quantized model is the student. Specifically, for the same input image, the student model needs to continuously minimize the discrepancy with the teacher model on the final super-resolution output. The loss for the final output can be written as

$$\mathcal{L}_O = \frac{1}{C_O H_O W_O} \|O - O_q\|_1 , \tag{4}$$

where $O$ and $O_q$ are the final outputs of the teacher and student models, $C_O$, $H_O$, and $W_O$ represent the number of output channels, height, and width, respectively. we adopt the L1 loss for the final output, as it tends to converge more easily compared to the L2 loss [33]. As the quantized model shares the same structure with the FP model and is quantized from the FP model, the student model also need to learn to extract the same feature of the teacher model, which can be written as

$$\mathcal{L}_F = \sum_i^N \frac{1}{C_i H_i W_i} \left\| \frac{F_i}{\|F_i\|_2} - \frac{F_{qi}}{\|F_{qi}\|_2} \right\|_2 , \tag{5}$$

where $F_i$ and $F_{qi}$ are the intermediate features of the teacher and student models respectively and $i$ is the index of the layer. In the field of super-resolution, there is a clear correspondence between the feature maps and the final reconstructed images, making training on feature maps crucial. since the quantized network

---

**Algorithm 1:** DOBI pipeline

**Data:** Data to be quantized $v$, the number of search point $K$, bit $b$
**Result:** Clip bound $l$, $u$
$l \leftarrow \min(v), u \leftarrow \max(v)$;
$min\_mse \leftarrow +\infty$;
**if** *v is symmetrical* **then**
    $\Delta l \leftarrow (\max(v) - \min(v))/2K$;
**else**
    $\Delta l \leftarrow 0$;
**end**
$\Delta u \leftarrow (\max(v) - \min(v))/2K$;
**while** $i \leq K$ **do**
    $l_i \leftarrow l + i \times \Delta l, u_i \leftarrow u + i \times \Delta u$;
    get $v_q$ based on Eq. (1);
    $mse \leftarrow \|v - v_q\|_2$;
    **if** $mse \leq min\_mse$ **then**
       $min\_mse \leftarrow mse$;
       $l\_best \leftarrow l_i, u\_best \leftarrow u_i$;
    **end**
**end**

---

and the full-precision network have identical structures, we do not need to add extra adaptation layers for feature distillation. The final loss function can be written as

$$\mathcal{L} = \mathcal{L}_O + \lambda \mathcal{L}_F, \tag{6}$$

where $\lambda$ is the co-efficient of $\mathcal{L}_F$. In the second stage, based on training optimization methods, the gap between the quantized model and the full-precision model will gradually decrease. The performance of the quantized model will progressively improve and eventually converge to the optimal range.

## 4 Experiments

### 4.1 Experimental settings

**Data and evaluation.** We use DF2K [40, 34] as the training data, which combines DIV2K [40] and Flickr2K [34], as utilized by most SR models. During training, since we employ a distillation training method, we do not need to use the high-resolution parts of the DF2K images. For validation, we use the Set5 [2] as the validation set. After selecting the best model, we tested it on five commonly used benchmarks in the SR field: Set5 [2], Set14 [45], B100 [36], Urban100 [18], and Manga109 [37]. On the benchmarks, we input low-resolution images into the quantized model to obtain reconstructed images, which we then compared with the high-resolution images to calculate the metrics. We do not use self-ensemble in the test stage as it increases the computational load eightfold, but the improvement in metrics is minimal The evaluation metrics we used are the most common metrics PSNR and SSIM [43], which are calculated on the Y channel (*i.e.*, luminance) of the YCbCr space.

**Implementation details.** We use SwinIR-light [32] as the backbone and provide its structure in the supplementary materials. We conduct comprehensive experiments with scale factors of 2, 3, and 4 and with 2, 3, and 4 bits, where Our hyperparameter settings remain consistent. During DOBI, we use a search step number of K=100, and the statistics of activations are obtained from 32 images in DF2K being randomly cropped to retain only $3\times64\times64$. During DQC, we use the Adam [23] optimizer with a learning rate of $1\times10^{-2}$, betas set to (0.9, 0.999), and a weight decay of 0. We employ CosineAnnealing [35] as the learning rate scheduler to stabilize the training process. Data augmentation is also performed. We randomly utilize rotation of 90°, 180°, and 270° and horizontal flips to augment the input image. The total iteration for training is 3,000 with batch size of 32. Our code is written with Python and PyTorch [38] and runs on an NVIDIA A800-80G GPU.

### 4.2 Comparison with state-of-the-art methods

The methods we compared include MinMax [22], Percentile [27], and the current SOTA post-quantization method in the super-resolution field, DBDC+Pac [41]. For a fair comparison, we report the performance of DBDC+Pac [41] on EDSR [33], as the authors performed detailed parameter adjustments and model training on EDSR. We directly used the results reported by the authors, recorded in the table as EDSR$^{\dagger}$. It should be noted that the EDSR method uses self-ensemble in the final test, which can improve performance to some extent but comes at the cost of 8 times the computational load. Additionally, we applied DBDC+Pac [41] to SwinIR-light [32], using the same hyperparameters as those set by the authors for EDSR, recorded in the table as DBDC+Pac [41]. The following are the quantitative and qualitative results of the comparison.

**Quantitative results.** Table 3 shows the extensive results of comparing different quantization methods with bit depths of 2, 3, and 4, as well as different scaling factors of $\times2$, $\times3$, and $\times4$.

DBDC+Pac [41] performs poorly mainly because **1.** The DBDC process requires manually specifying the clipping ratio, which significantly affects performance. **2.** DBDC does not prune weights, and the learning rate in the Pac process is too low, causing slow con-

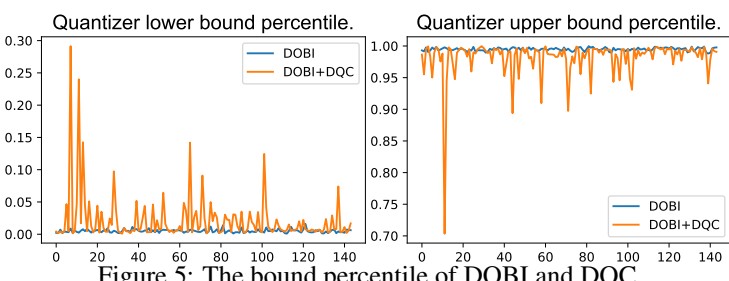

Figure 5: The bound percentile of DOBI and DQC.

vergence of weight quantizer parameters. However, both adverse factors are eliminated in our 2DQuant algorithm. When using only DOBI algorithm, our performance has already reached a level comparable to that of DBDC+Pac algorithms. Upon applying DQC, our performance experienced a remarkable and discernible enhancement, elevating it to new heights. In the case of $\times2$, 4-bit on Set5 and Urban100, DOBI has an improvement of 1.11dB and 0.39 dB compared to EDSR, while

Table 3: Quantitative comparison with SOTA methods. EDSR$^\dagger$ means applying DBDC+Pac [41] on CNN-based backbone EDSR [34]. Its results are cited from the paper [41].

| Method | Bit | Set5 (×2) PSNR↑ | SSIM↑ | Set14 (×2) PSNR↑ | SSIM↑ | B100 (×2) PSNR↑ | SSIM↑ | Urban100 (×2) PSNR↑ | SSIM↑ | Manga109 (×2) PSNR↑ | SSIM↑ |
|---|---|---|---|---|---|---|---|---|---|---|---|
| SwinIR-light [32] | 32 | 38.15 | 0.9611 | 33.86 | 0.9206 | 32.31 | 0.9012 | 32.76 | 0.9340 | 39.11 | 0.9781 |
| Bicubic | 32 | 32.25 | 0.9118 | 29.25 | 0.8406 | 28.68 | 0.8104 | 25.96 | 0.8088 | 29.17 | 0.9128 |
| MinMax [22] | 4 | 34.39 | 0.9202 | 30.55 | 0.8512 | 29.72 | 0.8409 | 28.40 | 0.8520 | 33.70 | 0.9411 |
| Percentile [27] | 4 | 37.37 | 0.9568 | 32.96 | 0.9113 | 31.61 | 0.8917 | 31.17 | 0.9180 | 37.19 | 0.9714 |
| EDSR$^\dagger$ [33, 41] | 4 | 36.33 | 0.9420 | 32.75 | 0.9040 | 31.48 | 0.8840 | 30.90 | 0.9130 | N/A | N/A |
| DBDC+Pac [41] | 4 | 37.18 | 0.9550 | 32.86 | 0.9106 | 31.56 | 0.8908 | 30.66 | 0.9110 | 36.76 | 0.9692 |
| DOBI (Ours) | 4 | 37.44 | 0.9568 | 33.15 | 0.9132 | 31.75 | 0.8937 | 31.29 | 0.9193 | 37.93 | 0.9743 |
| 2DQuant (Ours) | 4 | 37.87 | 0.9594 | 33.41 | 0.9161 | 32.02 | 0.8971 | 31.84 | 0.9251 | 38.31 | 0.9761 |
| MinMax [22] | 3 | 28.19 | 0.6961 | 26.40 | 0.6478 | 25.83 | 0.6225 | 25.19 | 0.6773 | 28.97 | 0.7740 |
| Percentile [27] | 3 | 34.37 | 0.9170 | 31.04 | 0.8646 | 29.82 | 0.8339 | 28.25 | 0.8417 | 33.43 | 0.9214 |
| DBDC+Pac [41] | 3 | 35.07 | 0.9350 | 31.52 | 0.8873 | 30.47 | 0.8665 | 28.44 | 0.8709 | 34.01 | 0.9487 |
| DOBI (Ours) | 3 | 36.37 | 0.9496 | 32.33 | 0.9041 | 31.12 | 0.8836 | 29.65 | 0.8967 | 36.18 | 0.9661 |
| 2DQuant (Ours) | 3 | 37.32 | 0.9567 | 32.85 | 0.9106 | 31.60 | 0.8911 | 30.45 | 0.9086 | 37.24 | 0.9722 |
| MinMax [22] | 2 | 33.88 | 0.9185 | 30.81 | 0.8748 | 29.99 | 0.8535 | 27.48 | 0.8501 | 31.86 | 0.9306 |
| Percentile [27] | 2 | 30.82 | 0.8016 | 28.80 | 0.7616 | 27.95 | 0.7232 | 26.30 | 0.7378 | 30.37 | 0.8351 |
| DBDC+Pac [41] | 2 | 34.55 | 0.9386 | 31.12 | 0.8912 | 30.27 | 0.8706 | 27.63 | 0.8649 | 32.15 | 0.9467 |
| DOBI (Ours) | 2 | 35.25 | 0.9361 | 31.72 | 0.8917 | 30.62 | 0.8699 | 28.52 | 0.8727 | 34.65 | 0.9529 |
| 2DQuant (Ours) | 2 | 36.00 | 0.9497 | 31.98 | 0.9012 | 30.91 | 0.8810 | 28.62 | 0.8819 | 34.40 | 0.9602 |

| Method | Bit | Set5 (×3) PSNR↑ | SSIM↑ | Set14 (×3) PSNR↑ | SSIM↑ | B100 (×3) PSNR↑ | SSIM↑ | Urban100 (×3) PSNR↑ | SSIM↑ | Manga109 (×3) PSNR↑ | SSIM↑ |
|---|---|---|---|---|---|---|---|---|---|---|---|
| SwinIR-light [32] | 32 | 34.63 | 0.9290 | 30.54 | 0.8464 | 29.20 | 0.8082 | 28.66 | 0.8624 | 33.99 | 0.9478 |
| Bicubic | 32 | 29.54 | 0.8516 | 27.04 | 0.7551 | 26.78 | 0.7187 | 24.00 | 0.7144 | 26.16 | 0.8384 |
| MinMax [22] | 4 | 31.66 | 0.8784 | 28.17 | 0.7641 | 27.19 | 0.7257 | 25.60 | 0.7485 | 29.98 | 0.8854 |
| Percentile [27] | 4 | 33.34 | 0.9137 | 29.61 | 0.8275 | 28.49 | 0.7899 | 27.06 | 0.8242 | 32.10 | 0.9303 |
| DBDC+Pac [41] | 4 | 33.42 | 0.9143 | 29.69 | 0.8261 | 28.51 | 0.7869 | 27.05 | 0.8217 | 31.89 | 0.9274 |
| DOBI (Ours) | 4 | 33.78 | 0.9200 | 29.87 | 0.8338 | 28.72 | 0.7970 | 27.53 | 0.8391 | 32.57 | 0.9367 |
| 2DQuant (Ours) | 4 | 34.06 | 0.9231 | 30.12 | 0.8374 | 28.89 | 0.7988 | 27.69 | 0.8405 | 32.88 | 0.9389 |
| MinMax [22] | 3 | 26.01 | 0.6260 | 23.41 | 0.4944 | 22.46 | 0.4182 | 21.70 | 0.4730 | 24.68 | 0.6224 |
| Percentile [27] | 3 | 30.91 | 0.8426 | 28.02 | 0.7545 | 27.23 | 0.7183 | 25.32 | 0.7349 | 29.43 | 0.8537 |
| DBDC+Pac [41] | 3 | 30.91 | 0.8445 | 28.02 | 0.7538 | 26.99 | 0.6937 | 25.10 | 0.7122 | 28.84 | 0.8403 |
| DOBI (Ours) | 3 | 32.85 | 0.9075 | 29.33 | 0.8200 | 28.27 | 0.7820 | 26.36 | 0.8036 | 31.14 | 0.9178 |
| 2DQuant (Ours) | 3 | 33.24 | 0.9135 | 29.56 | 0.8255 | 28.50 | 0.7873 | 26.65 | 0.8116 | 31.46 | 0.9235 |
| MinMax [22] | 2 | 26.05 | 0.5827 | 24.74 | 0.5302 | 24.42 | 0.4973 | 22.87 | 0.5155 | 24.66 | 0.5652 |
| Percentile [27] | 2 | 25.30 | 0.5677 | 23.60 | 0.4890 | 23.77 | 0.4751 | 22.33 | 0.4965 | 24.65 | 0.5882 |
| DBDC+Pac [41] | 2 | 29.96 | 0.8254 | 27.53 | 0.7507 | 27.05 | 0.7136 | 24.57 | 0.7117 | 27.23 | 0.8213 |
| DOBI (Ours) | 2 | 30.54 | 0.8321 | 27.74 | 0.7312 | 26.69 | 0.6643 | 24.80 | 0.6797 | 28.18 | 0.7993 |
| 2DQuant (Ours) | 2 | 31.62 | 0.8887 | 28.54 | 0.8038 | 27.85 | 0.7679 | 25.30 | 0.7685 | 28.46 | 0.8814 |

| Method | Bit | Set5 (×4) PSNR↑ | SSIM↑ | Set14 (×4) PSNR↑ | SSIM↑ | B100 (×4) PSNR↑ | SSIM↑ | Urban100 (×4) PSNR↑ | SSIM↑ | Manga109 (×4) PSNR↑ | SSIM↑ |
|---|---|---|---|---|---|---|---|---|---|---|---|
| SwinIR-light [32] | 32 | 32.45 | 0.8976 | 28.77 | 0.7858 | 27.69 | 0.7406 | 26.48 | 0.7980 | 30.92 | 0.9150 |
| Bicubic | 32 | 27.56 | 0.7896 | 25.51 | 0.6820 | 25.54 | 0.6466 | 22.68 | 0.6352 | 24.19 | 0.7670 |
| MinMax [22] | 4 | 28.63 | 0.7891 | 25.73 | 0.6657 | 25.10 | 0.6061 | 23.07 | 0.6216 | 26.97 | 0.8104 |
| Percentile [27] | 4 | 30.64 | 0.8679 | 27.61 | 0.7563 | 26.96 | 0.7151 | 24.96 | 0.7479 | 28.78 | 0.8803 |
| EDSR$^\dagger$ [33, 41] | 4 | 31.20 | 0.8670 | 27.98 | 0.7600 | 27.09 | 0.7140 | 25.56 | 0.7640 | N/A | N/A |
| DBDC+Pac [41] | 4 | 30.74 | 0.8609 | 27.66 | 0.7526 | 26.97 | 0.7104 | 24.94 | 0.7369 | 28.52 | 0.8697 |
| DOBI (Ours) | 4 | 31.10 | 0.8770 | 28.03 | 0.7672 | 27.18 | 0.7237 | 25.43 | 0.7631 | 29.31 | 0.8916 |
| 2DQuant (Ours) | 4 | 31.77 | 0.8867 | 28.30 | 0.7733 | 27.37 | 0.7278 | 25.71 | 0.7712 | 29.71 | 0.8972 |
| MinMax [22] | 3 | 19.41 | 0.3385 | 18.35 | 0.2549 | 18.79 | 0.2434 | 17.88 | 0.2825 | 19.13 | 0.3097 |
| Percentile [27] | 3 | 27.55 | 0.7270 | 25.15 | 0.6043 | 24.45 | 0.5333 | 22.80 | 0.5833 | 26.15 | 0.7569 |
| DBDC+Pac [41] | 3 | 27.91 | 0.7250 | 25.86 | 0.6451 | 25.65 | 0.6239 | 23.45 | 0.6249 | 26.03 | 0.7321 |
| DOBI (Ours) | 3 | 29.59 | 0.8237 | 26.87 | 0.7156 | 26.24 | 0.6735 | 24.17 | 0.6880 | 27.62 | 0.8349 |
| 2DQuant (Ours) | 3 | 30.90 | 0.8704 | 27.75 | 0.7571 | 26.99 | 0.7126 | 24.85 | 0.7355 | 28.21 | 0.8683 |
| MinMax [22] | 2 | 23.96 | 0.4950 | 22.92 | 0.4407 | 22.70 | 0.3943 | 21.16 | 0.4053 | 22.94 | 0.5178 |
| Percentile [27] | 2 | 23.03 | 0.4772 | 22.12 | 0.4059 | 21.83 | 0.3816 | 20.45 | 0.3951 | 20.88 | 0.3948 |
| DBDC+Pac [41] | 2 | 25.01 | 0.5554 | 23.82 | 0.4995 | 23.64 | 0.4544 | 21.84 | 0.4631 | 23.63 | 0.5854 |
| DOBI (Ours) | 2 | 28.82 | 0.7699 | 26.46 | 0.6804 | 25.97 | 0.6319 | 23.67 | 0.6407 | 26.32 | 0.7718 |
| 2DQuant (Ours) | 2 | 29.53 | 0.8372 | 26.86 | 0.7322 | 26.46 | 0.6927 | 23.84 | 0.6912 | 26.07 | 0.8163 |

2DQuant has an improvement of 0.69 dB and 1.18 dB compared to the SOTA method. All these results indicate that our two-stage PTQ method can effectively mitigate the degradation caused by quantization and ensure the quality of the reconstructed images.

Figure 5 shows the bound percentile of DOBI searching and DQC. Overall, the bound of DQC is tighter as the values around the zero point enjoy greater importance. Besides, the shallow layers'

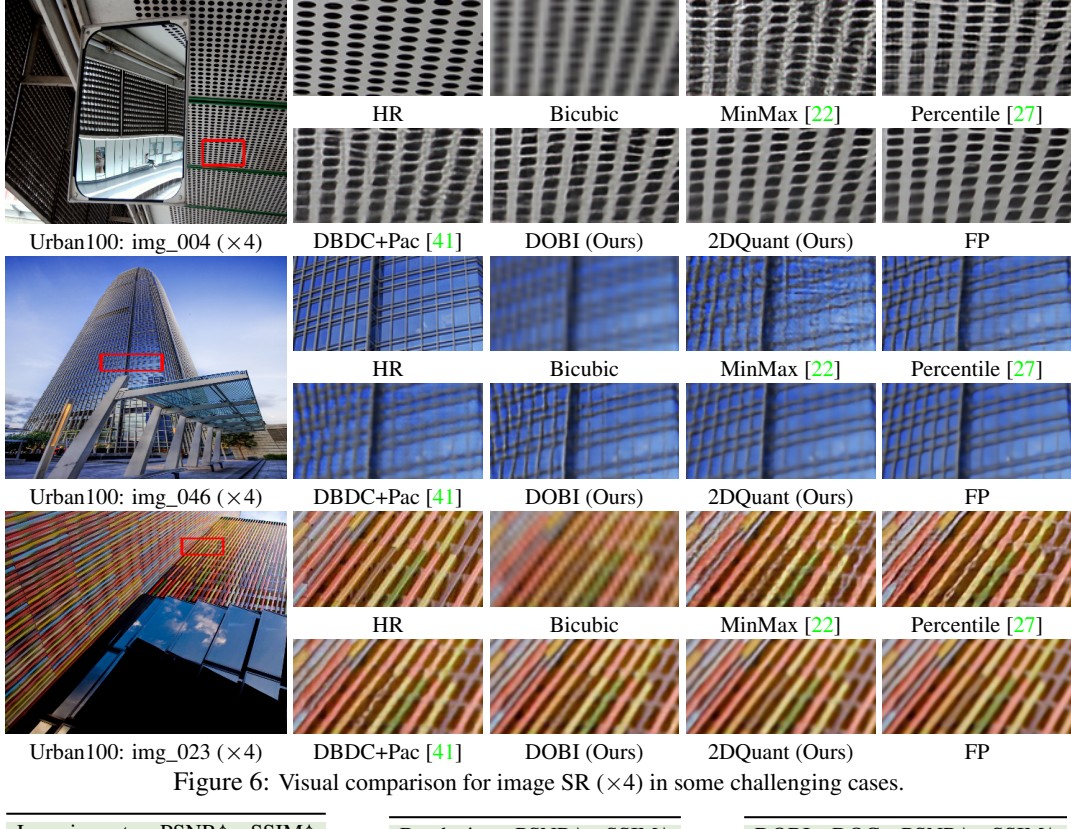

Figure 6: Visual comparison for image SR (×4) in some challenging cases.

| Learning rate | PSNR↑ | SSIM↑ |
|---|---|---|
| $10^{-1}$ | 37.82 | 0.9594 |
| $10^{-2}$ | 37.87 | 0.9594 |
| $10^{-3}$ | 37.78 | 0.9592 |
| $10^{-4}$ | 37.74 | 0.9587 |

(a) Learning rate

| Batch size | PSNR↑ | SSIM↑ |
|---|---|---|
| 4 | 37.82 | 0.9594 |
| 8 | 37.83 | 0.9594 |
| 16 | 37.84 | 0.9593 |
| 32 | 37.87 | 0.9594 |

(b) Batch size

| DOBI | DQC | PSNR↑ | SSIM↑ |
|---|---|---|---|
|  |  | 34.39 | 0.9202 |
| ✓ |  | 37.44 | 0.9568 |
|  | ✓ | 37.32 | 0.9563 |
| ✓ | ✓ | 37.87 | 0.9594 |

(c) DOBI and DQC

Table 4: Ablation studies. The models are trained on DIV2K and Flickr2K, and tested on Set5 (×2).

bounds vary more significantly due to the elevated significance of these layers within the neural network. Detailedly, the bound for the second MLP fully connected layer's weight in Layer 0 Block 1 only remains 46% data in its range. It has the second-highest lower bound percentile and the smallest upper bound percentile among the network. Its percentiles are 0.2401 and 0.7035 respectively while its bound values are -0.062 and 0.047 and its distribution is visualized in Figure 4. In conclusion, only through task-oriented optimization of each bound at a fine-grained level can redundant information be maximally excluded and useful information be maximally retained.

**Qualitative results.** We show the visual comparison results for ×4 in Figure 6. Since quantized models are derived from full-precision models with information loss, their global performance will rarely exceed that of full-precision models. As seen in the three images for Minmax, after quantization, if no clipping is performed, the long tail effect will lead to a large number of useless bits, resulting in a significant amount of noise and repeated distorted patterns in the reconstructed images. In these challenging cases, our training method allows the model to retain edge information of objects better, preventing blurring and distorted effects. For example, in img_046 and img_023, we have the highest similarity to the full-precision model, while other methods show varying degrees of edge diffusion, significantly affecting image quality. Compared to the DBDC+Pac method, our DOBI and DQC allow for better representation of edge and texture information in the images and effectively avoid distortions and misalignments in the graphics. The visual results demonstrate that our proposed DQC is essential for improving performance in both metric and visual comparisons.

### 4.3 Ablation study

**Learning rate and batchsize.** We first study the performance variations of the model under different hyperparameters. From Tables 4a and 4b, it can be seen that our DQC enables the model to

converge within a range of outstanding performance for most learning rates and batch sizes. Due to the non-smooth impact of quantization parameters on the model, the quantized model is more prone to local optima compared to the full-precision model, resulting in a noticeable performance drop when the learning rate is too low. Additionally, as shown in Table 4b, the larger the batch size, the better the model's performance, and the smoother the convergence process. However, even with a smaller batch size, we can still achieve a performance of 37.82dB on Set5, indicating that our two-stage method has good robustness to different hyperparameters.

**DOBI and DQC.**   Moreover, we also study the impact of different stages on performance, with the results shown in Table 4c, from which we can draw the following conclusions: **Firstly**, the goal of DOBI is to minimize the value shift for weights and activations. Although it is not the task goal, it can still enjoy significant enhancement due to better bit representational ability. **Secondly**, DQC alone cannot achieve the optimization effect of DOBI. This is because the impact of quantizer parameters on model performance is oscillatory, and training alone is prone to converge to local optima. In contrast, search-based methods can naturally avoid local optima. So it's necessary to use results from the search-based method to initialize training-based method in PTQ. **Thirdly**, when DOBI and DQC are combined, namely our 2DQuant, the 4-bit quantized model has only a 0.28dB decrease on Set5 compared to the FP model, which maximally mitigates the accuracy loss caused by quantization.

## 5   Discussion

**Why our results surpass FP outcomes.**   While our method's performance metrics do not yet fully match those of full-precision models, visual results reveal a compelling advantage. As observed in image img_092 of Figure 1 of Urban100, our approach correctly identifies the direction of the stripes in the image. Whereas the full-precision model erroneously selects the wrong direction. This discrepancy arises because the lower-resolution image, affected by aliasing, creates an illusion of slanted stripes, misleading the FP model's reconstruction. This phenomenon demonstrates that our PTQ algorithm allows more accurate restored results in certain localized and challenging tasks without being misled. More examples are in the supplementary materials.
It suggests that full-precision models contain not only redundant knowledge but also incorrect information. The latter is hard to get rid of by training the FP model. Our quantization method can effectively reduce model parameters and computational demands while eliminating erroneous information, achieving multiple benefits simultaneously. This also suggests that the FP model doesn't represent the pinnacle of what a quantized model can achieve.

**Limitations.**   Despite achieving excellent results, this study still has some limitations. During the DOBI process, the data distribution of activations and weights is required to approximate a bell curve or exponential distribution; otherwise, the DOBI method cannot find the most suitable positions. Additionally, increasing the number of search points for a single tensor in MSE does not necessarily guarantee better performance. However, the second-stage training can somewhat alleviate this issue. Moreover, our method requires a calibration set; without which, the first-stage DOBI and the second-stage DQC cannot be carried out at all.

**Societal impacts.**   Our super-resolution quantization method effectively saves computational resources, facilitating the deployment of super-resolution models at the cutting edge

## 6   Conclusion

This paper studies the post-training quantization in the field of image super-resolution. We first conducted a detailed analysis of the data distribution of Transformer-based model in SR. These data exhibit a clear long-tail effect and symmetry and asymmetry coexisting effect. We designed 2DQuant, a dual-stage PTQ algorithms. In the first stage DOBI, we designed two different search strategies for the two different distributions. In the second stage DQC, we designed a distillation-based training method that let the quantized model learn from the FP model, minimizing the accuracy loss caused by quantization. Our 2DQuant can compress Transformer-based model to 4,3,2 bits with the compression ratio being $3.07\times$, $3.31\times$, and $3.60\times$ and speedup ratio being $3.99\times$, $4.47\times$, and $5.08\times$. No additional module is added so 2DQuant enjoys the theoretical upper limit of compression and speedup. Extensive experiments demonstrate that 2DQuant surpasses all existing PTQ methods in the field of SR and even surpasses the FP model in some challenging cases. In the future, recognizing the significant impact of the model on performance, we will conduct PTQ research on more advanced super-resolution models and attempt to deploy quantized super-resolution algorithms to actual photography tasks, providing a more detailed evaluation of the performance of PTQ algorithms.

## Acknowledgments and Disclosure of Funding

This work is supported by the National Natural Science Foundation of China (62141220, 62271414), Shanghai Municipal Science and Technology Major Project (2021SHZDZX0102), the Fundamental Research Funds for the Central Universities, Zhejiang Provincial Distinguish Young Science Foundation (LR23F010001), Zhejiang "Pioneer" and "Leading Goose" R&D Program (2024SDXHDX0006, 2024C03182), the Key Project of Westlake Institute for Optoelectronics (2023GD007), and Ningbo Science and Technology Bureau, "Science and Technology Yongjiang 2035" Key Technology Breakthrough Program (2024Z126).

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
