# OpenReview forum: "2DQuant: Low-bit Post-Training Quantization for Image Super-Resolution"
_NeurIPS.cc/2024/Conference — NeurIPS 2024 poster_

### Official Review · Reviewer_EXmU · 2024-06-26

**Soundness:** 3
**Presentation:** 3
**Contribution:** 3
**Rating:** 6
**Confidence:** 5

**Summary:**

The paper presents 2DQuant, an innovative low-bit post-training quantization technique for transformer-based image super-resolution that significantly advances the state-of-the-art by introducing a two-stage optimization process. This process includes a novel Distribution-Oriented Bound Initialization strategy and a Distillation Quantization Calibration method, resulting in exceptional performance with minimal loss of accuracy for SwinIR-light. The 2DQuant approach demonstrates the potential to compress and accelerate transformer-based super-resolution models effectively for edge deployment.

**Strengths:**

1. This paper is well-written and easy to understand.

2. Transformer-based SR is much popular in recent year, this method is much helpful for obtaining accurate quantized transformer-based SR models.

3. The experimental results and visualizations are sufficient.

**Weaknesses:**

1. The proposed DOB and DQC are much similar with DBDC and PaC, but only change weight compression to symmetric quantization, could you give more details about the differences?

2. The motivation in the introduction are confusing. Usually, the deterioration of self-attention in quantized transformer causes sever performance degradation instead of the unadaptable changes in weight and activation distributions, which is also appeared in CNN-based SR networks.

3. The selected baselines in this paper are rough and the experimental results are not convinced. This paper does not compare with transformer-based post-training quantization methods, such as PTQ4ViT[1], FQ-ViT[2], NoiseQuant[3] and RepQ-ViT[4], these methods are open-source, the author should apply these PTQ methods for super resolution for that the basic transformer blocks are not different for super resolution and image classification.

[1] Z. Liu et al. Post-training quantization for vision transformer. Advances in Neural Information Processing Systems, 34:28092–28103, 2021.

[2] Y. Lin et al. Fq-vit: Post-training quantization for fully quantized vision transformer. arXiv preprint arXiv:2111.13824, 2021.

[3] Y. Liu et al. Noisyquant: Noisy bias-enhanced post-training activation quantization for vision transformers. In Proceedings of the IEEE/CVF Conference on Computer Vision and Pattern Recognition, pp. 20321– 20330, 2023.

[4] Z. Li et al. Repq-vit: Scale reparameterization for post-training quantization of vision transformers. In Proceedings of the IEEE/CVF International Conference on Computer Vision, pp. 17227–17236, 2023.

**Questions:**

Please refer to the weaknesses above.

**Limitations:**

The authors have discussed the limitations in this paper.

---

> ### Author Rebuttal · Authors · 2024-08-07
>
> ## Reviewer4 EXmU
>
> > Q1:The proposed DOB and DQC are much similar with DBDC and PaC, but only change weight compression to symmetric quantization, could you give more details about the differences?
>
> A1:
> In fact, our DOBI and DQC are quite different from DBDC and Pac. The specific differences are as follows:
>
> - **DOBI vs. DODB:**
>     1. We perform DOBI on both weights and activations, while DODB only on weights.
>     2. We use a one-sided search for activations, while DODB does not involve this aspect.
>     3. Our search objective is to find the minimum MSE, which can be accelerated by GPU, whereas DODB aims to find the minimum interval containing the T-th parameter size, where T is a hyper-parameter.
>
> - **DQC vs. Pac:**
>     1. Granularity is different. **DQC** is performed on the four Transformer layers in the SwinIR architecture, which is the largest substructure. We have tried finer granularity, such as layer-wise (24 in total) or linear-wise (96 in total), and the results indicate that this granularity is the most suitable. Limited by the number of pages, this part of the content was not included in the paper. In contrast, **DODB** is performed on each residual block, totaling 32 blocks, slower and worse than DQC.
>
> > Q2:The motivation in the introduction are confusing. ...
>
> A2:We assume that you are referring to lines 60-62 in the original text. Our statement is closer to the surface, while your statement is indeed the essence. We will modify this part to be "Secondly, most of these methods can not adapt well to Transformer-based models because of the deterioration of self-attention in quantized transformer.".
>
> > Q3:The selected baselines in this paper are rough and the experimental results are not convinced. This paper does not compare with transformer-based post-training quantization methods, such as PTQ4ViT[1], FQ-ViT[2], NoiseQuant[3] and RepQ-ViT[4]...
>
> A3: Thank you for your suggestions! For the four articles you mentioned, we have completed comparative experiments with them. The results of the experiments are shown below and more results can be found in the attachment file in Author Rebuttal.
>
> | Method | Bit | Set5($\times 2$) |  | Set14($\times 2$) |  | B100($\times 2$) |  | Urban100($\times 2$) |  | Manga109($\times 2$) |  |
> |---|---|:---:|:---:|:---:|:---:|:---:|:---:|:---:|:---:|:---:|:---:|
> |  |  | PSNR$\uparrow$ | SSIM$\uparrow$ | PSNR$\uparrow$ | SSIM$\uparrow$ | PSNR$\uparrow$ | SSIM$\uparrow$ | PSNR$\uparrow$ | SSIM$\uparrow$ | PSNR$\uparrow$ | SSIM$\uparrow$ |
> | Baseline | 32 | 38.15  | 0.961  | 33.86  | 0.921  | 32.31  | 0.901  | 32.76  | 0.934  | 39.11  | 0.978  |
> | Bicubic | 32 | 32.25  | 0.912  | 29.25  | 0.841  | 28.68  | 0.810  | 25.96  | 0.809  | 29.17  | 0.913  |
> | PTQ4ViT | 2 | 33.25  | 0.892  | 30.22  | 0.840  | 29.21  | 0.807  | 27.31  | 0.811  | 32.75  | 0.909  |
> | RepQ | 2 | 31.65  | 0.833  | 29.19  | 0.779  | 28.27  | 0.741  | 26.56  | 0.746  | 30.46  | 0.827  |
> | NoisyQuant | 2 | 30.13  | 0.762  | 28.80  | 0.756  | 28.26  | 0.742  | 26.68  | 0.763  | 30.40  | 0.820  |
> | Ours | 2 | 36.00  | 0.950  | 31.98  | 0.901  | 30.91  | 0.881  | 28.62  | 0.882  | 34.40  | 0.960  |
>
>
> | Method | Bit | Set5($\times 3$) |  | Set14($\times 3$) |  | B100($\times 3$) |  | Urban100($\times 3$) |  | Manga109($\times 3$) |  |
> |---|---|:---:|:---:|:---:|:---:|:---:|:---:|:---:|:---:|:---:|:---:|
> |  |  | PSNR$\uparrow$ | SSIM$\uparrow$ | PSNR$\uparrow$ | SSIM$\uparrow$ | PSNR$\uparrow$ | SSIM$\uparrow$ | PSNR$\uparrow$ | SSIM$\uparrow$ | PSNR$\uparrow$ | SSIM$\uparrow$ |
> | Baseline | 32 | 34.63  | 0.929  | 30.54  | 0.846  | 29.20  | 0.808  | 28.66  | 0.862  | 33.99  | 0.948  |
> | Bicubic | 32 | 29.54  | 0.852  | 27.04  | 0.755  | 26.78  | 0.719  | 24.00  | 0.714  | 26.16  | 0.838  |
> | PTQ4ViT | 2 | 29.96  | 0.790  | 27.36  | 0.700  | 26.74  | 0.659  | 24.56  | 0.646  | 27.37  | 0.739  |
> | RepQ | 2 | 27.32  | 0.648  | 25.63  | 0.592  | 25.44  | 0.565  | 23.42  | 0.558  | 24.51  | 0.572  |
> | NoisyQuant | 2 | 27.53  | 0.664  | 25.77  | 0.595  | 25.37  | 0.561  | 23.59  | 0.574  | 26.03  | 0.663  |
> | Ours | 2 | 31.62  | 0.889  | 28.54  | 0.804  | 27.85  | 0.768  | 25.30  | 0.768  | 28.46  | 0.881  |
>
>
> | Method | Bit | Set5($\times 4$) |  | Set14($\times 4$) |  | B100($\times 4$) |  | Urban100($\times 4$) |  | Manga109($\times 4$) |  |
> |---|---|:---:|:---:|:---:|:---:|:---:|:---:|:---:|:---:|:---:|:---:|
> |  |  | PSNR$\uparrow$ | SSIM$\uparrow$ | PSNR$\uparrow$ | SSIM$\uparrow$ | PSNR$\uparrow$ | SSIM$\uparrow$ | PSNR$\uparrow$ | SSIM$\uparrow$ | PSNR$\uparrow$ | SSIM$\uparrow$ |
> | Baseline | 32 | 32.45  | 0.898  | 28.77  | 0.786  | 27.69  | 0.741  | 26.48  | 0.798  | 30.92  | 0.915  |
> | Bicubic | 32 | 27.56  | 0.790  | 25.51  | 0.682  | 25.54  | 0.647  | 22.68  | 0.635  | 24.19  | 0.767  |
> | PTQ4ViT | 2 | 27.23  | 0.670  | 25.38  | 0.591  | 25.15  | 0.562  | 22.94  | 0.559  | 24.66  | 0.613  |
> | RepQ | 2 | 25.55  | 0.583  | 23.54  | 0.475  | 23.30  | 0.430  | 21.62  | 0.449  | 23.60  | 0.556  |
> | NoisyQuant | 2 | 25.94  | 0.586  | 24.33  | 0.507  | 24.16  | 0.472  | 22.32  | 0.484  | 23.82  | 0.540  |
> | Ours | 2 | 29.53  | 0.837  | 26.86  | 0.732  | 26.47  | 0.693  | 23.84  | 0.691  | 26.07  | 0.816  |

---

> ### Comment · Reviewer_EXmU · 2024-08-12
>
> Thank you for the response and additional experiments.  The specific differences about 2DQuant and PTQ4SR are clearly stated, which further shows that the differences between these two methods are marginal. DOBI on both weights and activations, one-side searching，new search object and different granularity are just engineering optimisations, the core distributions are still based on PTQ4SR. Although 2DQuant gets much better performance than baselines, the academic contributions of this paper are negligible.  And I agree with Reviewer DoGX that this paper use the A+B approach to exaggerate the contribution. So I decided to keep my score.

---

> > ### Author Response · Authors · 2024-08-13
> >
> > Dear Reviewer EXmU,
> >
> > We appreciate your constructive reviews on the additional experiments and the positive feedback on our performance. We would like to continue addressing your concerns and resolving any misunderstandings.
> >
> > > Q1: The specific differences about 2DQuant and PTQ4SR ... are marginal. DOBI on both weights and activations, one-side searching，new search object and different granularity are just engineering optimisations, the core distributions are still based on PTQ4SR.
> >
> > A1: We clarify that the only similarity between our DOBI and PTQ4SR is the use of a search algorithm, while all other aspects are different. If these differences are merely considered engineering optimizations, then under this assumption, most search-based PTQ methods would lack any innovative aspects, which is clearly unrealistic.
> >
> > > Q2: Although 2DQuant gets much better performance than baselines, the academic contributions of this paper are negligible.
> >
> > Our 2DQuant shows an increase in PSNR of up to 4.52 dB on Set5 (×2) compared with PTQ4SR. This increase is significant in the SR field. How can such a substantial improvement be considered negligible? The SR field urgently requires lightweight and high-performance DNN-based SR models. Our method demonstrates that a DNN-based SR model with 2-bit quantization can surpass Bicubic, which is the standard for practical use. Therefore, our method can greatly accelerate the deployment of SR models in the real world, representing a significant contribution to the SR field.
> >
> > > Q3: ...this paper uses the A+B approach to exaggerate the contribution.
> >
> > Could you please specify how we have exaggerated our contributions? We believe that your request for additional experiments is intended to further highlight and strengthen our contributions. The additional experiment results show that 2DQuant surpasses all other PTQ methods you requested for comparison, as well as the state-of-the-art PTQ4SR, confirming that we have not exaggerated our contributions.
> >
> > We value your feedback and have addressed each of your concerns in detail. The significant performance improvements demonstrated by 2DQuant, combined with its novel approach and impact on real-world deployment, affirm its substantial contribution to the SR field. We hope this clarification resolves any misunderstandings and further underscores the value of our work.
> >
> > Thank you for your consideration.
> >
> > Sincerely,
> >
> > Authors

---

> ### Comment · Reviewer_EXmU · 2024-08-14
>
> Thanks for the detailed explanation, I think I misunderstood a bit about the contributions of this paper. 2DQuant is much useful for accelerating the deployment of transformer-based image super resolution. This method is particularly beneficial to reduce the quantization error of  post-softmax and post-GELU feature maps in transformer architectures. After reviewing the latest response, I decided to raise my score to 6 (weak accept).

---

> > ### Author Response · Authors · 2024-08-14
> >
> > Dear Reviewer EXmU,
> >
> > We extend our sincerest gratitude for your thoughtful and constructive feedback on our 2DQuant. Your recognition of the innovative aspects of our work, particularly the utility of 2DQuant in enhancing the deployment of transformer-based image super-resolution models, is **greatly appreciated**. We are honored by your decision to raise your score to a "weak accept", reflecting your positive view of our research's contribution to model quantization and SR fields.
> >
> > **Thank you again for your time and the positive impact your suggestions have had on our work.**
> >
> > Sincerely,
> >
> > Authors

---

### Official Review · Reviewer_DoGX · 2024-07-08

**Soundness:** 2
**Presentation:** 2
**Contribution:** 1
**Rating:** 3
**Confidence:** 5

**Summary:**

This paper introduces a novel two-stage post-training quantization (PTQ) method aimed at compressing image super-resolution (SR) models for efficient deployment on edge devices. The authors address the challenge of accuracy degradation in low-bit quantization by proposing the 2DQuant method, which combines Distribution-Oriented Bound Initialization (DOBI) and Distillation Quantization Calibration (DQC) to achieve efficient and accurate super-resolution under low-bit quantization.

**Strengths:**

1. The writing of this paper is very clear.
2. The figures in the paper are simple and easy to understand.
3. This paper conducted some experiments to verify the effectiveness of the proposed method.

**Weaknesses:**

1. The contribution lacks novelty.
This paper does not bring any new insights. For example, the distribution of weights and activations is well studied by previous works[1][2][3]. Some papers have used searchable upper/lower bounds for quantization [2] (although these methods are based on QAT, the essence of QAT and PTQ is not different, with only slight differences in training strategies) and NO references were given in this manuscript. The proposed distillation approach is a widely used routine operation in quantization tasks [1][2][3][4][5][6].

2.  Experiments are insufficient.
The PTQ for transformers has been extensively studied in many other papers (e.g. [4][5]), but it seems that the authors have not compared it with these methods.

3.  Experimental settings are not consistent.
Only 100 images are used in [7], which is inconsistent with this paper.


References:
[1] Li, Huixia et al. “PAMS: Quantized Super-Resolution via Parameterized Max Scale.” ArXiv abs/2011.04212 (2020): n. pag.

[2] Zhong, Yunshan et al. “Dynamic Dual Trainable Bounds for Ultra-low Precision Super-Resolution Networks.” European Conference on Computer Vision (2022).

[3] Hong, Chee and Kyoung Mu Lee. “Overcoming Distribution Mismatch in Quantizing Image Super-Resolution Networks.” ArXiv abs/2307.13337 (2023): n. pag.

[4] Li, Yanjing et al. “Q-ViT: Accurate and Fully Quantized Low-bit Vision Transformer.” ArXiv abs/2210.06707 (2022): n. pag.

[5] Liu, Shi et al. “Oscillation-free Quantization for Low-bit Vision Transformers.” International Conference on Machine Learning (2023).

[6] Tu, Zhaopeng et al. “Toward Accurate Post-Training Quantization for Image Super Resolution.” 2023 IEEE/CVF Conference on Computer Vision and Pattern Recognition (CVPR) (2023): 5856-5865.

**Questions:**

See weaknesses.

**Limitations:**

Yes.

---

> ### Author Rebuttal · Authors · 2024-08-07
>
> > Q1a: The contribution lacks novelty... For example, the distribution of weights and activations is well studied by previous works.
>
> A1a: First, the study of distribution is not our core contribution or innovation, it is one of our experimental observations. While our main contributions are 1) the proposed 2DQuant, which utilizes DOBI and DQC to quantize Transformer-based model. 2) 2DQuant achieves 3.60x compression ration, 5.08x speedup ratio, and 4.52dB increasement in Set5 compared with SOTA[6]. Second, [1][2][3] focus on CNN-based models, such as ResNet, EDSR, and RDN networks, but our paper targets SwinIR, a Transformer-based model. There are significant differences between CNN networks and Transformer-based networks in terms of both architecture and model performance, so the direct application of these CNN-based insights/techniques on Transformer-based networks is not straightforward and even impossible.
>
> > Q1b: Some papers have used searchable upper/lower bounds for quantization (although these methods are based on QAT, the essence of QAT and PTQ is no different, with only slight differences in training strategies) and NO references were given in this manuscript.
>
> A1b:**First**, PTQ and QAT are two different concepts. PTQ stands for Post-Training Quantization, while QAT stands for Quantization-Aware Training. The former uses a pre-trained neural network without needing to update its parameters, only updating the quantizer parameters. In contrast, the latter typically uses an untrained neural network and updates both the neural network and quantizer parameters. From the perspective of memory usage, PTQ mainly requires storing the parameter values and their gradients (for backpropagation). The space required to store gradients is consistent with the space required for parameters. In addition to this, QAT needs to store the optimizer states for all parameters due to the necessity of updating the neural network's parameters. For example, Adam (used in [2]) requires saving momentum and squared gradients for each parameter, which has the same shape as the parameters. Therefore, for the same model, the memory requirement for QAT is typically twice that of PTQ, which is not a slight difference.
>
> **Second**, we are not a survey paper, so we do not need to cite all relevant articles. We have already cited similar works (like [6]), and not citing [2] is not a serious issue. To further enhance the quality of the paper, we will include a discussion of this article in the Related Work section.
>
> > Q1c: The proposed distillation approach is a widely used routine operation in quantization tasks.
>
> A1c: Although we both use a distillation method, there are significant differences in our specific implementations.
> 1. We only distill the quantizer parameters and do not need to distill the model parameters, which greatly reduces memory requirements and speeds up the algorithm.
> 2. The granularity of distillation is different. We target the largest substructure of the Transformer, which is the Transformer Layer. In SwinIR-light, there are a total of four Transformer layers, each containing six Transformer blocks. In contrast, [1-6] distills the residual blocks, of which there are 32 in EDSR—eight times more than ours. Therefore, our loss function computation will be faster and the distillation approach is different.
>
> > Q2:Experiments are insufficient...
>
> A2: Both [4] and [5] are QAT methods, training the model from scratch, while our proposed algorithm is a PTQ method. Comparing between PTQ method with QAT method is unfair as QAT often performs better than PTQ because of the fine-tuning of model parameters. A fairer comparison between [4][5] and our paper is applying their methods to optimize the quantizer parameters but freeze the model parameters. The result is shown below and more is in Author Rebuttal's attachment file. The additional experiments show our robustness and advantages.
>
> |Method|Bit|Set5(x4)||Set14(x4)||B100(x4)||Urban100(x4)||Manga109(x4)||
> |-|-|:-:|:-:|:-:|:-:|:-:|:-:|:-:|:-:|:-:|:-:|
> |||PSNR|SSIM|PSNR|SSIM|PSNR|SSIM|PSNR|SSIM|PSNR|SSIM|
> |OFQ*|3|30.16|0.854|27.26|0.746|26.73|0.705|24.25|0.714|26.93|0.843|
> |Ours|3|30.91|0.870|27.75|0.757|26.99|0.713|24.85|0.736|28.21|0.868|
> |OFQ*|2|29.15|0.827|26.59|0.725|26.33|0.688|23.59|0.680|25.61|0.803|
> |Ours|2|29.53|0.837|26.86|0.732|26.47|0.693|23.84|0.691|26.07|0.816|
>
> > Q3:Experimental settings are not consistent. Only 100 images are used in [7], which is inconsistent with this paper.
>
> A3: Our paper mentions using 32 images in line 228. **First**, we and [6] are two independent papers, so there is no need to adopt the same settings. **Second**, we achieved better performance with fewer images, which is our advantage instead of weakness. **Finally**, the results of 100 images are shown below, more can be seen in Author Rebuttal.
>
> |Set Size|Bit|Set5(x4)||Set14(x4)||B100(x4)||Urban100(x4)||Manga109(x4)||
> |-|-|:-:|:-:|:-:|:-:|:-:|:-:|:-:|:-:|:-:|:-:|
> |||PSNR|SSIM|PSNR|SSIM|PSNR|SSIM|PSNR|SSIM|PSNR|SSIM|
> |32|3|30.91|0.870|27.75|0.757|26.99|0.713|24.85|0.736|28.21|0.868|
> |100|3|30.94|0.870|27.79|0.757|27.02|0.713|24.90|0.737|28.22|0.867|
> |32|2|29.53|0.837|26.86|0.732|26.47|0.693|23.84|0.691|26.07|0.816|
> |100|2|29.82|0.843|27.03|0.736|26.57|0.696|24.01|0.698|26.40|0.823|
>
> References: [1] Li, Huixia et al. “PAMS: Quantized Super-Resolution via Parameterized Max Scale.” ECCV (2020).
>
> [2] Zhong, Yunshan et al. “Dynamic Dual Trainable Bounds for Ultra-low Precision Super-Resolution Networks.” ECCV (2022).
>
> [3] Hong, Chee and Kyoung Mu Lee. “Overcoming Distribution Mismatch in Quantizing Image Super-Resolution Networks.” ECCV (2024).
>
> [4] Li, Yanjing et al. “Q-ViT: Accurate and Fully Quantized Low-bit Vision Transformer.” NIPS (2022)
>
> [5] Liu, Shi et al. “Oscillation-free Quantization for Low-bit Vision Transformers.” ICML (2023).
>
> [6] Tu, Zhaopeng et al. “Toward Accurate Post-Training Quantization for Image Super Resolution.” CVPR (2023).

---

> > ### Comment · Reviewer_DoGX · 2024-08-08
> >
> > The author's rebuttal partially solved my problem. The authors mentioned that 'not urbanization [2] is not a serious issue.' In fact, PTQ and QAT are only different in the training process, and there is no difference in quantization essence. Since there are already highly similar quantization approaches in previous works, the authors should first consider clarifying the contribution of this paper during their writing, rather than attempting to use the A+B approach to avoid citation and exaggerate the contribution. So the final rating would be 3.

---

> > > ### Author Response · Authors · 2024-08-08
> > >
> > > Dear Reviewer DoGX,
> > >
> > > We appreciate your constructive reviews and positive feedback on our work, 2DQuant. We would like to continue addressing your concerns and resolving any misunderstandings.
> > >
> > > > Q1: The author's rebuttal partially solved my problem.
> > >
> > > First, you mentioned that we partially solved your problem. We would like to know what remains unresolved. We will do our best to address it. Please let us know the specifics.
> > >
> > > > Q2: In fact, PTQ and QAT are only different in the training process, and there is no difference in quantization essence.
> > >
> > > Second, you emphasize once again that QAT shares the same essence with PTQ. We do not agree with this point due to the differences in training cost, performance degradation, and task-specific designs. Nevertheless, we have provided detailed experimental results of [2]. We also mentioned in the rebuttal that we will cite [1] in our reversion. Our method outperforms [2] when following the PTQ settings. We think we have reached an agreement in this issue.
> > >
> > > > Q3: Since there are already highly similar quantization approaches in previous works...
> > >
> > > Third, we do not think the previous works are "highly similar" to ours. **IF so, how could our method significantly outperform other methods in the 2-bit senario?** The experimental results demonstrate that our approach is distinct from previous ones and **certainly not a simple combination of A and B**.
> > >
> > > > Q4: ...rather than attempting to use the A+B approach to avoid citation and exaggerate the contribution.
> > >
> > > Additionally, we must clarify that **we do not avoid citation or exaggerate the contribution**. As mentioned earlier, we will cite [1] in the reversion. Furthermore, the experimental results, not we, demonstrate the contributions, leaving no room for exaggeration.
> > >
> > > > Q5: So the final rating would be 3.
> > >
> > > We noticed that you scored **3** at first, but shortly after the beginning of the rebuttal, you revised it to **2** without explanation. After addressing your concerns and making efforts to resolve them, you reverted your rating back to **3**. We'd like to know why you decided to make the initial change. Could you please clarify the reason behind your initial change? Kindly list any unresolved issues so we can further improve the work.
> > >
> > > Thank you for your continued feedback and constructive engagement with our work. We look forward to resolving any remaining concerns and continuing the discussion to further improve our research.
> > >
> > > Best Regards,
> > >
> > > Authors
> > >
> > > [1] Zhong, Yunshan et al. “Dynamic Dual Trainable Bounds for Ultra-low Precision Super-Resolution Networks.” ECCV (2022).
> > >
> > > [2] Liu, Shi et al. “Oscillation-free Quantization for Low-bit Vision Transformers.” ICML (2023).

---

> > > > ### Author Response · Authors · 2024-08-14
> > > >
> > > > Dear Reviewer DoGX,
> > > >
> > > > We hope this message finds you well.
> > > >
> > > > In our preceding rebuttal, we made every effort to address your concerns and dispel any misunderstandings. Below is a summary of the issues we have resolved:
> > > >
> > > > - We have restated our primary contributions and clarified the difference between the observed distribution and those of prior studies.
> > > > - We have elucidated the reasons for our initial omission of reference [1] and have now included it in our revised manuscript.
> > > > - We have delineated the differences between DQC and previous distillation methodologies.
> > > > - We have conducted additional experiments, comparing our approach with five other quantization methods, thereby demonstrating our superior performance in the 2-bit scenario.
> > > > - We have explained the reason behind our choice of different calibration set sizes and have provided evidence of even better results when using the same set size.
> > > > - We have expounded on the divergences between our proposed PTQ method and other QAT approaches.
> > > > - We have clarified that we have neither avoided citations nor exaggerated our contributions.
> > > >
> > > > As the deadline for the rebuttal approaches, we have noticed that **Reviewer EXmU has acknowledged the merits of our work**, and we are eager to learn if there are any remaining concerns from your side that have not yet been addressed, to which we will provide further detailed explanations and clarification.
> > > >
> > > > We would like to express our gratitude once again for the time you have invested in reviewing our work and for the constructive feedback you have offered.
> > > >
> > > > Sincerely,
> > > >
> > > > Authors
> > > >
> > > > References:
> > > > [1] Zhong, Yunshan et al. “Dynamic Dual Trainable Bounds for Ultra-low Precision Super-Resolution Networks.” ECCV (2022).

---

> ### Comment · Reviewer_DoGX · 2024-08-14
>
> Dear Authors,
>
> I hope this message finds you well.
>
> My concerns still revolve around the similarity between the core idea of this article and [1]. Although you mentioned the differences in training details between PTQ and QAT (such as the parameters updating during backward pass) and the contribution of your method to SwinIR (which I have never denied), the academic contribution of this paper is still not convincing. The difference between SwinIR and CNN models lies in the model structure, rather than the constituent modules of the model. Therefore, applying existing quantization methods to new model architectures in this paper has a certain amount of engineering contribution rather than academic contribution. If the paper considers contribution from the perspective of engineering quantity, should deployment on practical reasoning frameworks become a necessary factor? In conclusion, I choose to maintain my score.
>
> Sincerely,
>
> Reviewer DoGX
>
> References: [1] Zhong, Yunshan et al. “Dynamic Dual Trainable Bounds for Ultra-low Precision Super-Resolution Networks.” ECCV (2022).

---

> > ### Author Response · Authors · 2024-08-14
> >
> > Dear Reviewer DoGX,
> >
> > Thank you for your quick response and for outlining the concern that remains unresolved.
> >
> > > Q1: My concerns still revolve around the similarity between the core idea of this article and [1]
> >
> > A1: The main contributions of [1] are 1) A layer-wise quantizer with trainable upper and lower bounds, 2) A dynamic gate controller to adaptively adjust the upper and lower bounds at runtime. We differentiate ourselves from [1] in the following aspects:
> >
> > - **Quantizer.** We have indeed utilized the same quantize. But this is widely used and is not our main contribution.
> > - **Different loss functions.** The granularity of the loss functions during training differs. Our DQC is conducted on the four Transformer layers within the SwinIR architecture, which represents the largest substructure. Experimentation with finer granularity, such as layer-wise (24 in total) or linear-wise (96 in total), has shown that this level of granularity is most optimal. Due to page limitations, this part of the content was omitted from the paper. In contrast, [1] applies quantization to each residual block, totaling 32 blocks in EDSR, which is slower and less effective than DQC.
> > - **No additional modules.** [1] employs a __Dynamic gate controller__, necessitating additional floating-point (FP) modules to obtain dynamic bounds, thereby increasing the inference cost. We, on the other hand, use entirely static quantizers and do not require any additional modules, achieving the theoretical optimal speedup ratio for quantization.
> > - **Initial values from DOBI.** [1] uses the percentile method to assign initial values, which is highly sensitive to the hyperparameters and thus requires a time-consuming search for the best parameters. Failure to do so can lead to **model collapse**, as reported in their GitHub repository issue 3. This is particularly problematic for Transformer-based models, where the self-attention modules contain many more activations. The consequence is that all elements in the attention map become the same value after quantization. Therefore, it is imperative to apply our efficient DOBI method for initial value assignment. The experimental results in Table 3 demonstrate that DOBI alone can achieve SOTA performance.
> >
> > Additionally, we would like to bring to your attention a **critical issue** in [1]'s implementation. They report the quantization method as per-layer quantization. But, in their github repo files(line 327-349 in model/quant\_ops.py in github repo \"zysxmu/DDTB\"), they actually uses different bounds for different channels in one activation tensor. This implement is a typical error for per-channel quantization. This is not appliable in real world GPU and will greatly **increase** the model performance. And the correct approach to quantization involves using a single pair of clipping bounds for activations and different pairs for different convolutional kernels, which is also noted in [2]. We hope you can focus on **correct** quantization approaches instead of one with implement error.
> >
> >
> > We trust these clarifications address your concerns and further highlight the unique contributions of our work.
> >
> > Sincerely,
> >
> > Authors
> >
> > References:
> >
> > [1] Zhong, Yunshan et al. "Dynamic Dual Trainable Bounds for Ultra-low Precision Super-Resolution Networks." ECCV (2022).
> >
> > [2] Markus Nagel et al. "A White Paper on Neural Network Quantization." arXiv:2106.08295.

---

### Official Review · Reviewer_Xp5V · 2024-07-08

**Soundness:** 3
**Presentation:** 3
**Contribution:** 4
**Rating:** 8
**Confidence:** 4

**Summary:**

This paper present a practical post-training quantization method for SR transformer, namely 2DQuant. The 2dquant mainly rely one two techniques, the first is Distribution-Oriented Bound Initialization (DOBI) determine the quantization range initially, and the second one is distillation Quantization calibration (DQC) to further finetune the quantizer for accurate quantization. Both of the proposed two techniques significantly improving the performance of PTQ for SR transformer, and the quantization can even be pushed to as low as 2-bit without retraining.

**Strengths:**

1. The proposed method is a PTQ method without full retraining instead of the usually studied quantization-aware training methods for SR tasks, which can be seen as resource-saving in real applications. Considering the targeted architectures are SR transformers is always used as well-pretrained models, the PTQ pipeline is significant and practical.

2. The proposed quantization method for SR transformer are clear and effective. DOBI allows the optimization of quantization ranges start from a statistical search-based optimal, and then use DQC to fully optimized the quantizer. It’s brings improvements of both weight and activation quantization, especially for the activation with dynamic distribution that requires more robust and powerful quantizer.

3. Experiments show that the proposed method achieve SOTA results on ultra-low 2-4 bit Sr networks, which allows SR transformer enjoys very good efficiency with little accuracy loss. And the visualization also show the good performance of the networks quantized by the proposed methods.

4. The paper is easy to follow and with good writing and presentation, and with comprehensive visualization for the distribution of both weights and distributions.

**Weaknesses:**

1. As presented in section 3, the authors use fake quantization during the design of the method, but didn’t mention if it can be replaced by the real quant, which can be implemented and bring real acceleration in hardware. The author should discuss if the proposed method can achieve real quantized inference when deployment.

2. The algorithm of DQC is missed and make the description not very clear, I suggest to add it or merge it to the algorithm of DOBI. And Figure 3 not highlight the application of the proposed techniques, which is also suggested to revise.

3. Some details of writing should be improved and polished carefully, e.g., the subtitles of section 2 and 3, some explain of the proposed equations are missing.

**Questions:**

Please respond for the issues raised in weaknesses part.

**Limitations:**

The authors have discussed the limitation in their paper.

---

> ### Author Rebuttal · Authors · 2024-08-07
>
> > Q1:As presented in section 3, the authors use fake quantization during the design of the method, but didn’t mention if it can be replaced by the real quant, which can be implemented and bring real acceleration in hardware. The author should discuss if the proposed method can achieve real quantized inference when deployment.
>
> A1: Fake quantization is a widely used technique in model quantization. Its purpose is to simulate the precision loss caused by quantization. Using fake quantization, we can obtain the model parameters of the quantizer. With the quantizer's parameters, we can deploy and test the model using common quantization techniques. We omitted this part in the article because the quantization method we adopted is already very mature in this regard and our speedup ratios are obtained from real deployment.
>
> Besides, the speedup ratio reported in lines 20-22 of our paper is the actual （real quant） deployment speedup ratio——2DQuant achieves an increase in PSNR of up to 4.52 dB on Set5 (×2) compared to SOTA when quantized to 2 bits, along with a 3.60× compression ratio and a 5.08× speedup ratio.
>
> > Q2a:The algorithm of DQC is missed and make the description not very clear, I suggest to add it or merge it to the algorithm of DOBI.
>
> A2a: Thank you for your suggestion. The nature of DQC is distillation between the FP model and the quantized model, which optimizes the parameters of the quantizers. We will merge this part of the algorithm into the DOBI algorithm.
>
> > Q2b: And Figure 3 not highlight the application of the proposed techniques, which is also suggested to revise.
>
> A2b: You are correct.
>
> Figure 3 is not closely related to the proposed techniques. However, it is still very necessary to present Figure 3. We included Figure 3 to show that in the Transformer block, we have quantized all the computationally intensive parts, such as BMM, FC, etc, some of which are often forgotten to be quantized. Other parts, such as Softmax and Layernorm, have relatively small computational loads, but quantizing these parts would have a significant negative impact on the model.
>
> > Q3: Some details of writing should be improved and polished carefully, e.g., the subtitles of section 2 and 3, some explain of the proposed equations are missing.
>
> A3: Thank you very much for your suggestions on writing details. We have already made the changes you mentioned in the arXiv version.

---

> > ### Comment · Reviewer_Xp5V · 2024-08-12
> > **After rebuttal**
> >
> > Thank you for the response. After reviewing the rebuttal, I can confirm that all of my concerns have been fully addressed.

---

### Official Review · Reviewer_6qcS · 2024-07-09

**Soundness:** 3
**Presentation:** 3
**Contribution:** 4
**Rating:** 7
**Confidence:** 5

**Summary:**

The authors propose a low-bit post-training quantization (PTQ) method, 2DQuant, for image super-resolution. 2DQuant is a dual-stage low-bit PTQ method. They first investigate the weight and activations. They propose Distribution-Oriented Bound Initialization (DOBI) by using different searching strategies to get a coarse bound for quantizers. They further propose Distillation Quantization Calibration (DQC) to refine the quantizer parameters with a distillation approach. They provide extensive experiments on different bits and scaling factors. When compared with SOTA methods, there proposed methods achieve superior performance with 3.6X compression ratio and 5.08X speedup ratio in the 2-bit case.

**Strengths:**

1)The authors explore PTQ to Transformer based image super-resolution (SR), which is one of the first works in this research field. This topic is also very practical and is favored in real-world applications.

2)The authors investigate this topic with some observations and visualizations. For example, the visualizations in Figures 4, 5, 7-11, are inspiring.

3)They propose distribution-oriented Bound Initialization (DOBI) to minimize the value heterogenization between quantized and the FP models. They DOBI to search for quantizer parameters by employing customized search strategies for different distributions to balance speed and accuracy.

4)They propose Distillation Quantization Calibration (DQC) to adjust each bound to its best position finely. DQC is used to ensure the quantized model align with the FP model on the calibration set.

5)In the ablation study, the authors investigate the effect of some key components, like DOBI and DQC.

6)In the main comparisons with other SOTA methods, the proposed method obtain the best performance and visual results. I believe this low-bit quantization work in Transformer achieves excellent permance and has good potential in future research.

7)The authors give detailed derivation of their backward gradient propagation formula, which makes the method more convincing.

**Weaknesses:**

The authors did not give results about inference time. The quatization can reduce params and ops largely. Then, how about the acceleration when deploying the quantized models into device?

Some parts in the experiments are confusing. For example, in Table 3, the results of EDSR confuse me somehow. I am not very sure how the EDSR results are obtained. The original EDSR is full-precision model. Here, the authors report its quantized version. The baseline in Table 3 is not very clear. Please clarify them.

In Table 4 (b), when the batch size becomes larger, the performance changes slightly. This is not very consistent with the common observation, where larger batch size usually improves performance.

The writing can be further improved. Some details should be given attention and revised. For example, Section 3.2, the first letter should be capital.

**Questions:**

In Figure 1, the proposed method performs even better than the full-precision (FP) version. What are the potential reasons behind this observation?

Can this method be applied to diffusion based image SR models? If so, please give some key ideas. It will be much better if the authors can provide some preliminary results.

In Table 3, what is the model size for the original EDSR? I do not which version of EDSR did the authors use in the paper.

What is the motivation by providing two versions, i.e., DOBI (ours) and 2DQuant (ours), in Table 3?

------------After Rebuttal and  Discussions----------
After rebuttal and disscussions, my concerns have been well solved.

**Limitations:**

The authors have discussion the limitation in their paper.

Flag For Ethics Review: No ethics review needed.

---

> ### Author Rebuttal · Authors · 2024-08-07
>
> > Q1: ... how about the acceleration when deploying the quantized models into device?
>
> A1: In the field of quantization research, when we apply the same quantization method to the same neural network module, the speedup ratio of the model does not change due to variations in the quantizer parameters. Therefore, the algorithm typically focuses on how to better optimize the quantizer parameters to improve model performance after applying such quantization methods.
>
> Besides, the speedup ratio reported in lines 20-22 of our paper is the actual deployment speedup ratio——2DQuant achieves an increase in PSNR of up to 4.52 dB on Set5 (×2) compared to SOTA when quantized to 2 bits, along with a 3.60× compression ratio and a 5.08× speedup ratio.
>
> > Q2:.... I am not very sure how the EDSR results are obtained. The baseline in Table 3 is not very clear. Please clarify them.
>
> A2: We clarify that the EDSR in Table 3 does not refer to the original FP model. Its specific meaning is explained in lines 237-239 – it is the test result of the previous SOTA method DBDC+Pac on EDSR. It should be noted that the parameter size of EDSR is 172.36MB, while the parameter size of SwinIR-light is 3.42MB, which is about 2% of the former. As for the baseline in Table 3, it is the full-precision version of SwinIR-light.
>
> Our comparison with EDSR aims to illustrate that:
> 1. The best result in [1] was achieved on EDSR, and applying our quantization method on SwinIR-light performs better than their method on EDSR;
> 2. The choice of the quantized model itself is crucial. To achieve better quantization results, we cannot only experiment on classic models (as done in the [1]), because in terms of parameter size, cutting-edge models before quantization already outperform quantized classic models.
>
> > Q3:In Table 4 (b), when the batch size becomes larger, the performance changes slightly. This is not very consistent with the common observation, that larger batch size usually improves performance.
>
> A3:Thank you for your keen observation. The main reason for this issue lies in the fact that DOBI provides excellent quantizer parameters. In the SR field, the numerical evaluation metric usually adopted is PSNR. If we only look at PSNR, with the increase of batch size, our performance shows a slight improvement. The reason for this slight improvement is that our one-stage search algorithm, DOBI, has already provided quite optimal quantizer parameters, resulting in a very high starting performance for the model. At the same time, the initialization parameters provided by DOBI ensure that the model can achieve almost the same excellent results regardless of changes in batch size during subsequent training.
>
> I believe this phenomenon is beneficial because, even when GPU memory is limited, we can still achieve results comparable to those obtained with a large batch size by reducing the batch size and extending the training time for DQC.
>
> > Q4: The writing can be further improved. Some details should be given attention and revised. For example, Section 3.2, the first letter should be capital.
>
> A4: Thank you for your suggestions on writing details. We have revised in the Arxiv version.
>
> > Q5:In Figure 1, the proposed method performs even better than the full-precision (FP) version. What are the potential reasons behind this observation?
>
> A5:We clarify that the main reason is that FP models tend to overfit the training set, leading to weaker generalization performance on **some** test data. Model quantization might alleviate this overfitting. Although there might be an overall performance drop, the performance on certain data can be better than that of the FP model.
>
> > Can this method be applied to diffusion based image SR models? If so, please give some key ideas. It will be much better if the authors can provide some preliminary results.
>
> A6:Yes! The core components of diffusion are the denoising network and the sampler, with UNet being a common form of the denoising network. The computationally intensive parts of the UNet are the convolutional layers. We can use the DOBI and DQC methods mentioned in the article to perform two-stage quantization of the convolutional layer parameters.
>
> However, it is important to note that the generation process of diffusion models typically involves several steps, and the distribution of activations is usually associated with the time steps. Therefore, designing algorithms tailored to each time step might yield better results. For example, we can divide the time steps into different segments and apply different quantizers to the activations in different segments.
>
> > Q7:In Table 3, what is the model size for the original EDSR? I do not which version of EDSR did the authors use in the paper.
>
> A7: As mentioned in A2, the EDSR in Table 3 refers to the data in Table 3 of [1]. Details and parameters of the model used can be found in the following link at line 11 in the blob/master/src/model/edsr.py file in the github repo "sanghyun-son/EDSR-PyTorch". To our knowledge, this is the largest version of EDSR.
>
> > Q8: What is the motivation by providing two versions, i.e., DOBI (ours) and 2DQuant (ours), in Table 3?
>
> A8: DOBI is our one-stage algorithm while 2DQuant represents the final results of both the one-stage and two-stage processes. The reason we included both in Table 3 is to demonstrate that even just using DOBI, the model can exceed the SOTA performance on most datasets, and the two-stage algorithm can further enhance the model's performance.
>
> [1] Tu, Zhaopeng et al. “Toward Accurate Post-Training Quantization for Image Super Resolution.” 2023 IEEE/CVF Conference on Computer Vision and Pattern Recognition (CVPR) (2023): 5856-5865.

---

> > ### Comment · Reviewer_6qcS · 2024-08-14
> >
> > Thanks for your detailed responses, which have well addressed my concerns. I also go through other reviewer's comments, and recognize the contributions of  low-bit post-training quantization (PTQ) for image super-resolution with using different searching strategies to get a coarse bound for quantizers. Overall, I tend to keep my original score as 6.

---

### Author Rebuttal · Authors · 2024-08-07

Dear Reviewers and Area Chairs,

We appreciate all reviewers (R1-6qcS, R2-Xp5V, R3-DoGx, R4-EXmU) for the constructive reviews and positive feedback to our 2DQuant. Your expertise and insightful comments help us to further improve our paper.

We are pleased that:

- R1 and R2 acknowledge the practicality of our techniques.
- R1, R2, and R4 think our visualizations are inspiring, comprehensive, or sufficient.
- R1 and R2 recognize the impressive performance of our proposed DOBI and DQC.
- R2, R3, and R4 appreciate our writing as easy to follow or clear.

We have responded individually to each reviewer to address any concerns. Here, we offer a summary:

- We clarify the acceleration, deploying problem, fake quantization, and EDSR related problems.
- We discuss the reason that the Quant model outperforms FP model, the application of 2DQuant on the Diffusion model, the batch size ablation, and the motivation of table design.
- We explain the difference between our method and [7].
- We improve the writing, including motivation in the introduction, the subtitle of Section 2 and 3, and merging DQC into DOBI.
- We compare our approach with four PTQ methods, and one QAT method and provide analysis. The results are in the **attached PDF**.

Thanks again to all the reviewers and area chairs. We appreciate you taking the time to review our responses and hope to discuss further whether the issues have been resolved. If you need any clarification, please let us know.

Best Regards,

Authors

---

The attached PDF includes:

1. Table 1: Quantitative comparison with PTQ and QAT methods.
2. Table 2: Ablation on size of calibration set.

## Discussion of Table 1

Reviewer EXmU requests us to compare with more PTQ methods, such as PTQ4ViT[1], FQ-ViT[2], NoiseQuant[3], and RepQ-ViT[4]. Reviewer DoGX requests us to compare with more QAT methods, such as Q-ViT[5] and OFQ[6]. We have compared our methods with OFQ[6] in the PTQ fashion and all the PTQ methods. The results are shown in **Table 1 in the attached PDF file**. We fail to compare with Q-ViT[5] as their code assigns inappropriate initial values (see line 162-165 and line 188-193 in  \_quan\_base.py file in YanjingLi0202/Q-ViT) to the quantizers and the pre-trained model fails to perform Forward Propagation, let alone optimization.

For a fair comparison, we made necessary adjustments to their codes.

- FQ-ViT quantizes all modules in SwinIR, which certainly leads to worse performance compared with partly-quantized models. So we align its quantization scheme with ours, only quantize the linear and BMM modules in the Transformer block.
- NoiseQuant doesn't quantize the BMM part in the Transformer block and we fixed this bug.
- OFQ is a QAT method. For a fair comparison, we apply OFQ in our task but freeze the model parameters, only optimize the quantizer parameters.

We mark the highest with red and the second highest with blue. It's worth noting that our method **gains SOTA performance in 2bit and 3bit situations** and is _slightly_ lower than RepQ. But RepQ uses **per-channel** quantization while ours is per-tensor. And in one transformer block, RepQ needs **128** quantizers but 2DQuant only takes **16** quantizers. If switched to per-tensor quantization, RepQ's performance is lower than ours in all situations, which is not shown in Table 1 due to page limitation. Besides, FP-ViT adopts minmax optimizer, which leads to model collapse in 4bit. So it's not necessary to compare it with lower bits.


## Discussion of Table 2

Table 2 shows the result. With a larger calibration set, 2DQuant gains higher performance in all situations.

Reference:
[1] Z. Liu et al. “Post-training quantization for vision transformer.” NIPS (2021).

[2] Y. Lin et al. “Fq-vit: Post-training quantization for fully quantized vision transformer.” IJCAI (2022).

[3] Y. Liu et al. “Noisyquant: Noisy bias-enhanced post-training activation quantization for vision transformers.” CVPR (2023).

[4] Z. Li et al. “Repq-vit: Scale reparameterization for post-training quantization of vision transformers.” ICCV (2023).

[5] Li, Yanjing et al. “Q-ViT: Accurate and Fully Quantized Low-bit Vision Transformer.” NIPS (2022).

[6] Liu, Shi et al. “Oscillation-free Quantization for Low-bit Vision Transformers.” ICML (2023).

[7] Tu, Zhaopeng et al. “Toward Accurate Post-Training Quantization for Image Super Resolution.” CVPR (2023).

---

### Decision · Program_Chairs · 2024-09-25

**Decision:**

Accept (poster)

**Comment:**

This paper proposed a PTQ-based SR transformer. Relying on Distribution-Oriented Bound Initialization (DOBI) & distillation Quantization calibration (DQC), the method is able to achieve better compression(around 4x) and speed acceleration(around 5x).

Most reviewers appreciated the technical advancements leaning toward acceptance although one reviewer tends to reject it. 3 reviewers agreed the novelty of the work, but reviewer DoGX still has different opinion on that. Reviewer DoGX raised concerns about the novelty  (similarity between [ref1]), lack of baselines (i.e. PTQ-based Transformer) and need for demonstration about scalability. In my opinion, the authors resolved most of raised concerns in rebuttal phase. Reviewer DoGX is still unsatisfied especially on the novelty of the work, but this AC agreed the other 3 reviwers' perspective which are on consensus about certain differences from [ref1]. In my point of view, the paper demonstrated SR transformer with simulated quantization besides trainable ranges and successful results within PTQ schematic whereas [ref1] is based on QAT and demonstrated mostly on CNNs. Further, two reviewers (Xp5V & 6qcS) highlighted the importance of the paper (8 strong accept & 7 accept).

Therefore, I recommend the acceptance of the paper for publication.

[ref1] Zhong, Yunshan et al."Dynamic Dual Trainable Bounds for Ultra-low Precision Super-Resolution Networks." ECCV (2022).